# Materials and Life Science Experimental Facility at the Japan Proton Accelerator Research Complex IV: The Muon Facility

**Wataru Higemoto [1], Ryosuke Kadono [2],\*, Naritoshi Kawamura [2], Akihiro Koda [2], Kenji M. Kojima [2], Shunsuke Makimura [2], Shiro Matoba [2], Yasuhiro Miyake [2], Koichiro Shimomura [2] and Patrick Strasser [2]**

[1] J-PARC Center and Advanced Science Research Center, JAEA, Tokai, Ibaraki 319-1195, Japan; higemoto.wataru@jaea.go.jp

[2] J-PARC Center and Institute of Materials Structure Science, High Energy Accelerator Research Organization (KEK), Tsukuba, Ibaraki 305-0801, Japan; nari.kawamura@kek.jp (N.K.); coda@post.kek.jp (A.K.); kenji.kojima@kek.jp (K.M.K.); shunsuke.makimura@kek.jp (S.Mak.); shiro.matoba@kek.jp (S.Mat.); yasuhiro.miyake@kek.jp (Y.M.); koichiro.shimomura@kek.jp (K.S.); patrick.strasser@kek.jp (P.S.)

\* Correspondence: ryosuke.kadono@kek.jp; Tel.: +81-29-284-4896

**Abstract:** A muon experimental facility, known as the Muon Science Establishment (MUSE), is one of the user facilities at the Japan Proton Accelerator Research Complex, along with those for neutrons, hadrons, and neutrinos. The MUSE facility is integrated into the Materials and Life Science Facility building in which a high-energy proton beam that is shared with a neutron experiment facility delivers a variety of muon beams for research covering diverse scientific fields. In this review, we present the current status of MUSE, which is still in the process of being developed into its fully fledged form.

**Keywords:** positive muon; muonium; muon spin rotation; negative muon; muonic atom; muonic X-ray

## 1. Overview of the Muon Science Establishment

### 1.1. Muon Science Facility with Tandem-Type Production Graphite Target

The Materials and Life Science Facility (MLF) consists of neutron and muon science facilities that utilize a 3-GeV proton beam (beam power 1 MW, repetition rate 25 Hz). It was decided to use the muon production target in tandem with that for neutrons on a single proton beamline (see Figure 1) rather than constructing one for muons in a separate building with a dedicated proton beamline and beam dump. This resulted in the significant reduction of the total construction cost by sharing the beam between the neutron and muon facilities, thereby enabling the common use of utilities while avoiding the enormous work of beam dump construction, including high-level tritium water handling. The total beam loss induced by placing the muon production target is specified to be 10% or less. Although two muon-production targets made of graphite with thicknesses of 10 mm and 20 mm could be installed in the beamline upstream from the neutron target [1], as was originally planned, a single 20-mm-thick graphite target is currently utilized for muon production. The heat deposited into a 25-mm-diameter spot on this target by a 1-MW proton beam is estimated to be 3.9 kW.

In the earlier stage of development, an edge-cooled fixed (non-rotating) graphite target was adopted, because of the relative ease of handling and maintaining it. Since September 2008, the fixed graphite target has been utilized without any trouble. In 2014, to reduce the frequency of future

target replacement due to the relatively short lifetime expected for the fixed target in the 1-MW era, a rotating graphite target based on a working model developed at the Paul Scherrer Institute (PSI) was installed [2].

### 1.2. Building Structure and Maintenance

The MLF building consists of proton beamline tunnels (M1 and M2) and two wings for experimental halls (numbers 1 and 2, on the east and west sides, respectively). The tunnel structure is intended to contain radioactive products within the tunnel that may be emitted during maintenance of the neutron and/or muon targets. Since a certain fraction of the primary 3-GeV proton beam is scattered toward the neutron target, two sets of beam collimators (called "scrapers") are installed to prevent radiation damage to the beamline components, including the quadrupole magnets and beam ducts. Based on the experiences accumulated at the PSI in dealing with 1-MW proton beams, all of the beamline components were designed to enable remote handling from the maintenance area located 2.4 m above the beamline level [2].

### 1.3. Muon Beamlines

In Phase 1 of the original Japan Proton Accelerator Research Complex (J-PARC) construction plan, which was split into two parts, a superconducting decay/surface muon beamline (D-line) with a modest acceptance (about 40 msr) for pion capture was installed in hall No. 2, where both surface and decay muons can be delivered to experimental areas D1 and D2. A total surface muon ($\mu^+$) flux of $1.8 \times 10^6$/s was achieved using a 120-kW proton beam in 2009, surpassing the previous world-record flux realized at the RIKEN muon facility at the ISIS facility of the Rutherford Appleton Laboratory (RAL) in the UK [3]. A total surface muon flux of $1.5 \times 10^7$/s is anticipated to be achievable by utilizing the designed proton beam power of 1 MW.

Following the successful operation of the D-line, the "Super Omega" muon beamline (U-line) was installed, which consists of a large-acceptance rad-hard solenoid, a superconducting curved transport solenoid, and superconducting axial focusing magnets, which together produce an "ultra-slow muon" (USM) beam.

In experimental hall No. 1, a new surface muon beamline (S-line) dedicated to the material sciences and a "high-momentum" muon beamline (H-line) are under construction. The S-line is planned to have four experimental areas (S1–S4), and the beamline to area S1 was completed by 2016.

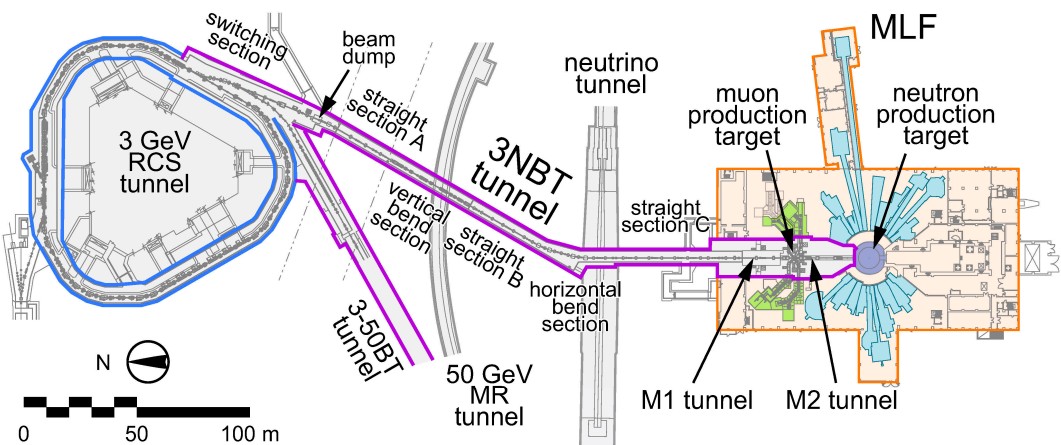

**Figure 1.** Schematic drawing of the proton beamline from the 3-GeV rapid cycling synchrotron (RCS) to the Materials and Life Science Facility (MLF).

## 2. Muon Target

### 2.1. Overview of the Muon Target

As shown in Figure 1, the proton-beamline tunnel runs through the center of the MLF building from north to south. The muon production target is located about 30 m upstream from the neutron production target. To prevent the diffusion of radioactive contamination generated around the muon target, the proton beamline tunnel is isolated from the experimental halls. The muon target system and front-end devices for the secondary muon beamlines are located in the proton beamline tunnel (M2). The muon target system consists of the target itself and the adjacent beam scrapers that prevent protons scattered by the target from reaching the downstream devices between the muon and neutron targets and causing radiation damage. Some of the components near the muon target are expected to be highly radioactive, making it impossible to perform hands-on maintenance. In the case of the muon target, the radiation dose rate is predicted to be 5 Sv/h after a year of 1-MW proton beam irradiation. Thus, these components are designed to be handled remotely from the maintenance area located 2.4 m above the beamline level and separated by 2-m-thick Fe shielding.

As described previously in Section 1.3, the highest flux pulsed muon beam in the world is produced by a 20-mm-thick graphite target (IG-430U, Toyotanso Co., Ltd., Osaka, Japan) [4], in which about 5% of a proton beam is spared for muon production. More specifically, 4% is lost due to reactions with C nuclei and 1% is for Coulomb scattering. After muons were first produced in September 2008, the proton beam intensity was gradually increased to 300 kW in January 2013 for stable operation [5,6]. The fixed muon target remained in service without replacement until May 2014.

Proton irradiation causes radiation damage to the material properties of graphite [7,8]. In particular, the effects on the dimensions of graphite are serious. The lifetime of the fixed target was estimated to be 1 year, based on a simulation of graphite irradiation with a 1-MW proton beam. Because the muon target is highly activated by proton irradiation, the used target must be handled remotely in a "hot cell", which requires considerable time, cost, and manpower [9]. Thus, it is crucial to extend the target lifetime to minimize the frequency with which the facility operation must be interrupted to replace it. To this end, the development of a rotating target, in which the radiation damage is spread across a wider area, began in 2008, simultaneously with the operation of the fixed target [10]. Since its installation into the proton beamline in September 2014, the rotating muon target has operated successfully without replacement.

### 2.2. Fixed Muon Target

The fixed muon target is made of 20-mm-thick, 70-mm-diameter isotropic graphite. The beam profile has a Gaussian distribution with a standard deviation of approximately 3.5 mm. The energy deposited by the 1-MW proton beam on the muon target has been estimated to be 3.9 kW using a particle transport simulation code (PHITS) [11,12]. Because the target is located in a vacuum chamber, the cooling water traveling through the stainless-steel tube must remove most of the heat embedded in the Cu frame. To absorb the thermal stress, a Ti layer is employed as an intermediate material between the graphite target and Cu frame [13]. Figure 2 shows a picture of the fixed muon target, taken from 12 m upstream along the proton beamline, where the thermocouples measuring the temperature of the Cu frame can be seen. Upon replacement of the fixed target with the rotating target in 2014, the fixed target had incurred no damage during proton beam operation for nearly 6 years.

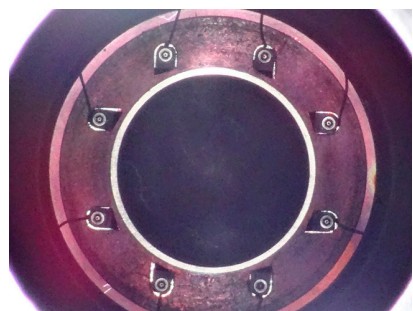

**Figure 2.** Picture of the fixed muon target, taken from 12 m upstream along the proton beamline (before upgrading to the current rotating target).

Radioactive damage induced by proton irradiation deteriorates the material properties of graphite, such as its thermal conductivity, strength, and dimensions. Dimensional changes of graphite could have particularly serious effects. These factors must be addressed when evaluating the target lifetime by considering three kinds of stresses. One type of stress is the residual (mostly compressive) stress induced by shrink-fitting during manufacturing to maintain high thermal conduction at the interfaces of different materials. The second type is thermal stress due to the temperature distribution caused by proton beam irradiation, where compressive stress is expected due to the restriction of the thermal expansion of the high-temperature beam spot by the surrounding low-temperature part. The thermal conductivity degradation due to proton irradiation must be considered when simulating this type of stress. The third type of stress is the tensile stress resulting from the dimensional changes after continuous beam irradiation for a certain time. The highly irradiated beam spot on the graphite disk will exhibit shrinkage under the constraints imposed by the surrounding part, where the accumulated stress is proportional to the operational time and hysteresis.

The simulation indicated that the theoretical lifetime of the graphite target was less than 1 year under 1-MW proton irradiation. To evaluate the practical lifetime of the currently used target, the detailed proton irradiation history, including the beam position on the target, which was altered at each operational time to disperse the irradiation damage, was carefully considered. The beam spot was concentrically moved away from the center in 4-mm increments to 8 different positions almost every 1000 h when the beam power was 200 kW and every 700 h when the beam power was 300 kW [14].

The fixed muon target was successfully replaced with the new rotating target in September 2014 [15]. After temporary storage for cooling in a storage pod, the highly radioactive used target was contained within a shielding vessel called a "transfer cask" for transport to the hot cell, where it was handled and replaced with the new target using the remote handling devices. The used target was cut into pieces by a special device in the hot cell to enable the storage volume available for the highly activated parts to be used as efficiently as possible.

Remote handling commissioning in the hot cell has been conducted almost once a year. Figure 3 shows pictures of the target chamber in which the muon target and stainless-steel shielding are inserted, the transfer cask during the replacement of the fixed target with the rotating target, and a mockup target loaded onto a remote-handling device in the hot cell, from left to right.

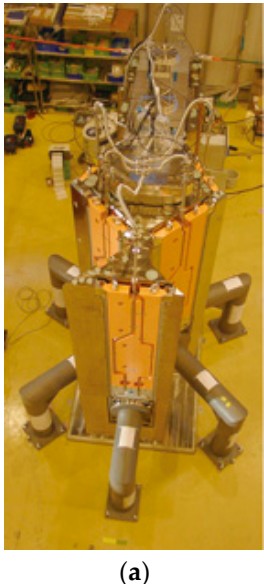 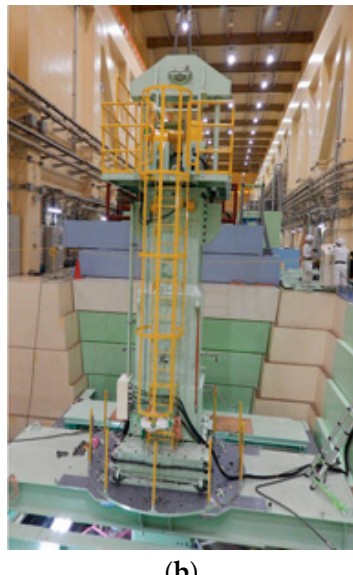 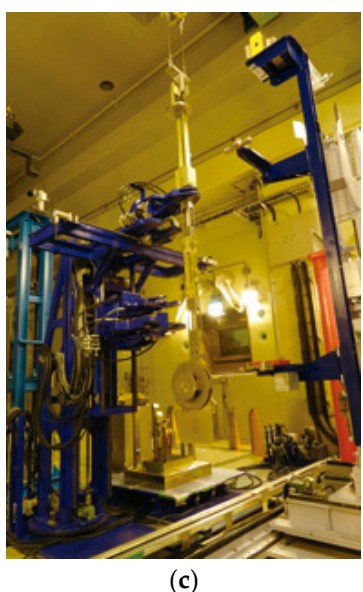

(**a**)           (**b**)           (**c**)

**Figure 3.** Pictures of (**a**) the target chamber, (**b**) the transfer cask, and (**c**) remote handling commissioning in the hot cell.

### 2.3. Rotating Muon Target

The new rotating target is designed to have a horizontal shaft parallel to the proton beam line to serve as a rotation axis. Because the motor used to rotate the target must be located 2.4 m above the beamline level to avoid exposure to high radiation doses, the rotating motion is transmitted to the horizontal shaft through a long vertical shaft and pair of bevel gears. Figure 4a shows a picture of the entire target assembly. As shown in Figure 4b, the rotating body is composed of a graphite wheel, a wheel support, and the horizontal shaft supported by two horizontal bearings. The two bearings are attached to a cooling jacket in which water piping is embedded. The temperatures of the graphite wheel and hottest bearing were estimated to be 940 K and 390 K, respectively. The target rotation speed was determined to be 15 rpm based on evaluation of the maximum temperature gradients inside the graphite wheel.

Because the lifetime of the graphite wheel under continuous rotation is estimated to be more than 30 years, the bearing lifetime will be the factor limiting the lifetime of the entire system. The bearings (supplied by JTEKT Co., Ltd., Osaka, Japan) [16] are located in a high vacuum of $10^{-4}$ Pa, at a high temperature of 390 K, and under a high radiation dose of 100 MGy/year. In these conditions, disulfide molybdenum ($MoS_2$) or Ag is generally used as a coating lubricant. We adopted a sintered *compact* of disulfide tungsten ($WS_2$) instead, because $WS_2$ coating is expected to have a much longer lifetime than $MoS_2$ or Ag coating (~110,000 h, or 22 years assuming proton beam operation for 5000 h/year).

Since the fixed target was replaced with the rotating target, no problems have developed over a period including 300–400 kW proton beam operation for 3 months, 500-kW operation for 1 month, and 600-kW operation for 1 h. Approximately $1 \times 10^{22}$ protons have been supplied to the rotating target, and the cumulative number of revolutions has reached 4.8 million.

During proton beam operation, the temperatures of the cooling jacket, horizontal shaft, and graphite are monitored. Because it is difficult to measure the temperatures of the rotating bodies directly, the temperature increases in the thermocouples due to thermal radiation are monitored. The temperature of the horizontal shaft is continuously monitored using thermocouples inserted into a hollow core in the center of the shaft. The thermocouples for target monitoring are isolated by thermal shielding and are irradiated only by the graphite disk. While the measured temperatures of the horizontal shaft were found to agree closely with the simulated results, those of the graphite disk exhibited a significant discrepancy whose cause is yet to be clarified. In any case, the effects of

heat generation due to proton beam irradiation must be carefully confirmed during actual operation. Table 1 compares the measured and simulated temperatures at typical beam powers.

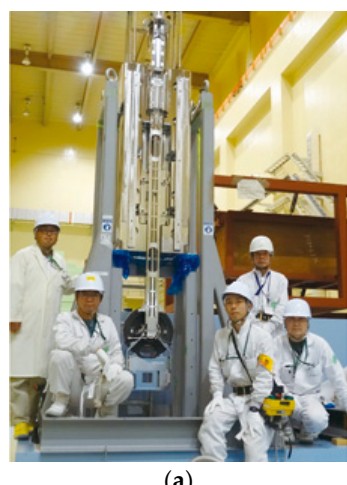
(**a**)

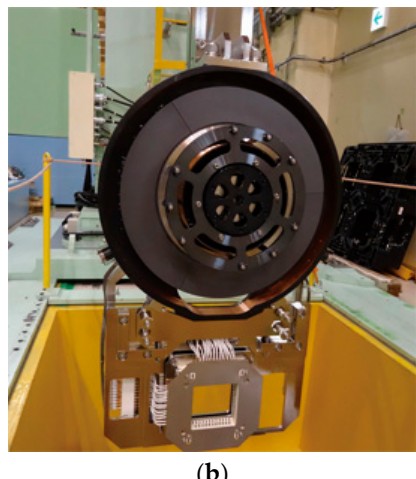
(**b**)

**Figure 4.** Pictures of (**a**) the rotating muon target assembly; and (**b**) the rotating muon target itself.

**Table 1.** Measured and simulated temperatures of the horizontal shaft and maximum simulated temperatures of the graphite at typical beam powers.

|  | 300 kW | 500 kW | 1 MW |
| --- | --- | --- | --- |
| Shaft Simulation | 71 °C | 84 °C | 115 °C |
| Shaft Measurement | 78 °C | 95 °C | - |
| Graphite Simulation | 400 °C | 475 °C | 620 °C |

*2.4. Scrapers*

Two scrapers (SCs, made of O-free Cu) are placed downstream from the rotating muon target in a series to reduce the proton beam halo. Each of them is approximately 300 mm wide, 700 mm high, and 700 mm long in the direction of the beam axis and has a circular bore along the beam axis with an inlet diameter of 74 mm that increases downstream. They are exposed to heat loads of about 19 kW during 1-MW operation due to the beam halo induced by proton scattering from the target. The cooling water pipes are embedded in the SCs using the hot isostatic press technique to minimize the thermal resistance at the interfaces between the pipes and Cu block. K-type thermocouples enclosed in stainless steel sheaths are mounted on the upstream and side surfaces to measure the SC temperatures.

We encountered a problem in that the SC temperatures measured by the thermocouples increased unexpectedly after upgrading to the rotating target. It was later determined that the local heating of the halo monitor mounted on the upstream surface of each SC was causing thermal radiation to the thermocouples that were directly facing hot spots. Therefore, the SCs were replaced with improved SCs without halo monitors, where thermocouples were attached behind the boring structure as shown in Figure 5, so that they would not be directly exposed to the thermal radiation from the upstream components, including the rotating target. For the replacement work, the radioactivity induced by the beam irradiation was evaluated using PHITS and a residual radioactivity calculation code, DCHAIN-SP. The radioactivity due to all of the radionuclides was determined to be about 20 TBq.

As of 2017, the new SCs are running smoothly with 150 kW operation. The temperature of the new SCs was measured to be 42 °C during 500-kW operation in the second half of 2015, which is close to the design value, suggesting that temperature measurement of the new SCs should be feasible even during 2-MW operation.

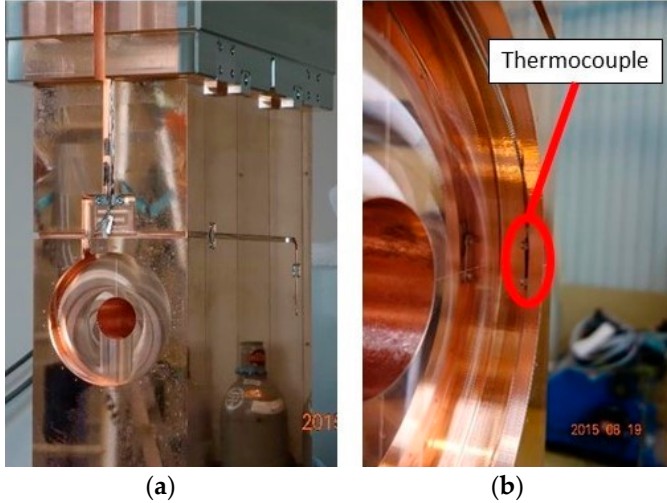

**Figure 5.** Pictures of (**a**) one of the new scrapers (SCs); and (**b**) the region near the thermocouple.

## 3. Muon Beamlines

Muons are emitted by pions generated via nuclear reactions in the graphite target, where two different modes are commonly used to extract them as secondary beams. The first type of secondary beam, called a "surface-muon" beam, is produced by collecting the muons emitted by the positive pions that come to a halt at the subsurface of the graphite target. The other type is called a "decay-muon" beam and is produced by collecting the muons from positive or negative pions that decay during flight (i.e., transport) along the beamline.

As shown in Figure 6, the muon-production target is surrounded by four muon beamlines (the D-, U-, S-, and H-lines) extending to the two experimental halls. The beamlines are designed to provide high-flux muon beams with different properties to meet the demands of a variety of muon experiments. Two of the beamlines form 60° angles with respect to the proton beamline (forward), and the others are at 135° (backward). The front-end parts of these beamlines are located within the proton beamline tunnel, leaving little room for future modification as a trade-off for radiation safety. The beamline construction proceeded successively from the D-line, to the U-line, and then to the S-line, which have been in operation since 2009, 2011, and 2015, respectively, while the H-line is still under construction.

The D-line can deliver both negatively and positively charged muons with momenta of up to 120 MeV/c, in addition to surface muons (~30 MeV/c) and so-called "cloud muons," where the latter are the muons generated by the lowest energy ("cloud"-like) pions emitted by the muon target. This high flexibility in terms of beam momentum/charge is suitable for serving a wide variety of user programs. The U-line is next to the D-line in hall No. 1 and is characterized by high muon acceptance, enabling the delivery of the highest flux surface-muon beam among the four secondary beamlines. This high-flux beam is used to generate a USM beam via the resonant ionization of thermal muonium by a laser [17]. The S-line is designed to transport surface muons to four experimental areas in experimental hall No. 2 to provide beams simultaneously to muon spin rotation (μSR) spectrometers using muon-kicker devices. The construction is currently complete up to the first experimental area (S1). The H-line is designed to meet the demands of particle physics experiments that require long-term occupancy of beamlines/experimental areas and therefore are incompatible with short-term user programs for the material sciences.

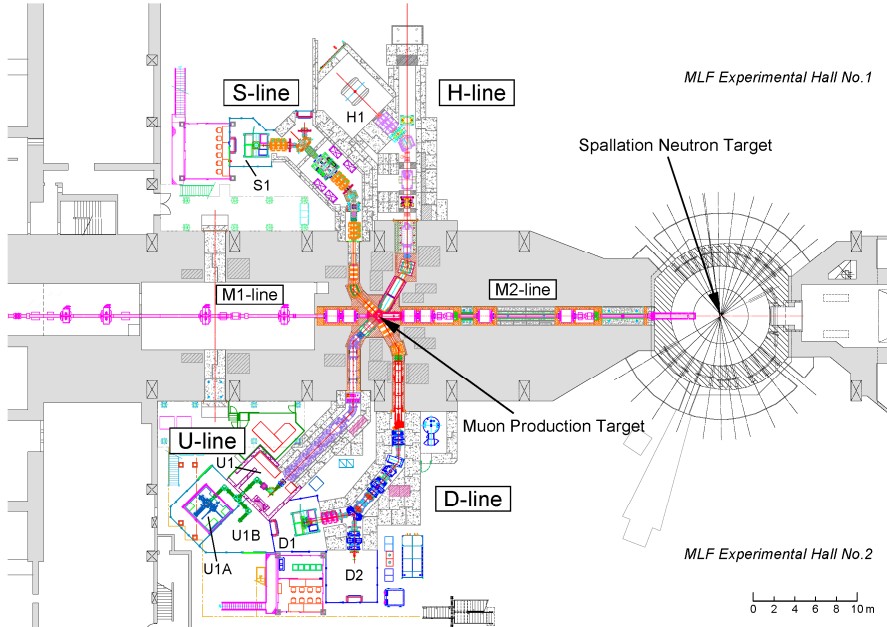

**Figure 6.** Floor plan of the muon beamlines (provisional for the H-line) in the MLF building. Four beamline tunnels, one each for the D-, U-, S-, and H-lines, surround the muon production target.

### 3.1. D-Line

The D-line is the first muon beamline to have been completed in the Muon Science Establishment (MUSE) facility [18], and it is currently employed for various kinds of muon experiments, including condensed matter physics and material science experiments using the μSR technique, nondestructive elemental analysis, and fundamental muon physics investigations. It can deliver surface (positive) muons with a total flux of $10^7$–$10^8$/s and decay positive/negative muons with momenta of up to 120 MeV/c and a flux of $10^6$–$10^7$/s at 60 MeV/c. Recently, a total surface muon flux of $4.5 \times 10^6$/s over a beam spot size of ø40 mm FWHM at the sample position was achieved during 300 kW operation. As shown in Figure 7, the entire beamline consists of three main sections: one for pion injection, a superconducting decay solenoid, and one for muon extraction. It has two branches downstream from the last bending magnet ("septum" magnet), each of which extends to an experimental area (D1 or D2). A DC separator ("Wien filter") placed after the second bend is used in combination with slits to separate muons from the positrons/electrons in the beam.

Similarly to the proton beam produced at ISIS-RAL, the beam provided by the rapid cycling synchrotron (RCS) at J-PARC has a double-pulsed structure in which each pulse has a width of ~100 ns; the pulses are separated by 600 ns and are delivered at a repetition rate of 25 Hz. Accordingly, the muon beam has nearly the same time structure as that at ISIS-RAL. To utilize the muon beam more efficiently, a muon-kicker system was installed to split the beam into two single-pulsed beams for simultaneous delivery to the two experimental areas, enabling two μSR experiments to be conducted in parallel [19]. The system consists of two magnetic kickers, two switchyard magnets, and a septum magnet. The switchyard magnets deflect the first muon pulse of a double-pulsed beam so that the beam is shifted to the right at the entrance of the septum magnet to be transported to area D1. Then, the kicker magnets deflect the second muon pulse in the opposite direction to inject it into the septum magnet on the left that leads to area D2. The rise time for kicker excitation is less than 300 ns with a flat top of about 300 ns, while the decay time is not critical since the following muon pulse arrives only after 40 ms. The system was designed for operation with a momentum of up to 60 MeV/c to deliver beams simultaneously to areas D1 and D2, while a double-pulsed beam can be delivered alternately to one of these areas with a momentum of up to 120 MeV/c.

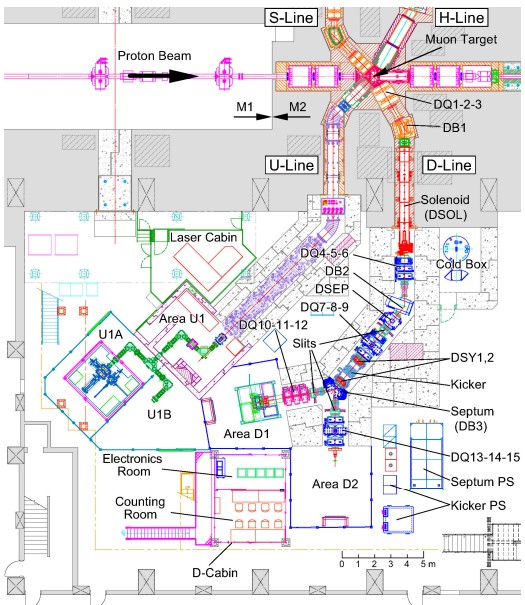

**Figure 7.** Layout of the D-line in MLF experimental hall No. 2. The key components are the superconducting decay solenoid that brings muons from the proton beam tunnel into the experimental hall, the kicker system that separates the double-pulsed muon beam, and the septum magnet that delivers muons to areas D1 and D2.

Until recently, the D-line relied on a superconducting solenoid magnet that had been in service for more than 30 years since its first installation at the former Booster Meson facility at KEK. It had a cold bore with an inner radius of ⌀12 cm and was equipped with thermal shield windows at both ends that prevented the transport of relatively low-momentum muons due to scattering at those windows. The solenoid was finally replaced in 2015 by a new one containing an inner warm bore with an inner radius of ⌀20 cm and no thermal windows. This replacement enabled the extraction of intense negative muons with momenta much lower than those that could be extracted previously. Preliminary beam commissioning results showed that there was more than a tenfold increase in the intensity of negative muons with momenta less than 20 MeV/c. This new development will be highly advantageous for experiments involving nondestructive elemental analysis of thin samples.

### 3.2. U-Line

The main purpose of the U-line is to deliver a high-flux surface muon beam for the production of intense USMs, where "ultra-slow" means a kinetic energy of $10^2$–$10^3$ eV, which is low enough to be stopped within a few nanometers from the surfaces of materials. Such muons will extend the scope of the µSR technique from bulk materials to thin films, near-surfaces, interfaces, and extremely small samples and will facilitate not only a wide variety of nano-science studies, but also novel 3D imaging with "ultra-slow muon microscopes" [20].

USMs are produced as a tertiary beam by the re-acceleration of thermal muons regenerated by the laser resonant ionization of muonium atoms evaporated from a hot W foil, a method that originated at the Meson Science Laboratory at KEK [21]. Since the yield of USMs relative to that of surface muons is ~$10^{-3}$, the muon flux available at the end of the U-line is crucial for the feasibility of the present method based on hot W foil.

The U-line, or "Super-Omega muon beamline," consists of three magnet devices, i.e., a normally conducting capture solenoid, a superconducting curved transport solenoid, and an axial focusing solenoid [22,23]. Muons with momenta of up to 45 MeV/c can be captured with a large acceptance solid angle (400 msr) of the front-end solenoid in tunnel M2, which faces the upstream side of the muon

target at 135°. The captured muons are then transported to the experimental hall by a superconducting curved transport solenoid and to area U1 by an axial focusing solenoid. Since the beamline consists entirely of solenoids, surface $\mu^+$ and cloud $\mu^-$ are transported simultaneously, which is inconvenient for most experiments. Muon charge selection is performed using two dipole coils installed in the straight section of the curved solenoid [24]. Recently, we achieved the highest pulsed muon flux in the world with a time-averaged surface-muon intensity of $6.4 \times 10^7$/s using a proton beam power of 212 kW, which is 20 times more intense than at the D-line and is equivalent to $3.0 \times 10^8$/s with the designed beam power of 1 MW.

The layout of the USM beamline is illustrated in Figure 8. In area U1, a high-flux muon beam irradiates a hot W foil placed in the muonium chamber, and laser beams passing near the foil surface ionize muonium atoms evaporated into the vacuum. To ionize the muonium atoms efficiently, a resonant ionization scheme involving the "1s–2p–unbound" transition and a pulsed nanosecond laser was adopted. A new state-of-the-art all solid-state laser system produces vacuum ultraviolet (VUV) light with a wavelength of 122.088 nm (Lyman-alpha), which is necessary to induce the 1s–2p transition [25,26]. After laser ionization, the free muons have a mean thermal kinetic energy of only 0.2 eV with 50% spin polarization. They are re-accelerated up to 30 keV and focused by an electrostatic lens for transport via a series of electric quadrupoles and electric bends to two experimental areas, U1A (for USM-μSR) and U1B (for further re-acceleration, microbeam μSR, muon transmission microscopy, etc.) [27]. In area U1A, a μSR spectrometer furnished with a load-lock system for sample exchange in an ultra-high vacuum is installed on a high-voltage platform (±30 kV) to enable USM implantation without causing discharge around the sample (see Section 4.2). During this stage of commissioning, a total USM flux of 42 $\mu^+$/s has been attained (February 2017), which is higher than that achieved previously at RAL (20 $\mu^+$/s). This flux was measured at an intermediate position after the first electric bend with a beam size of approximately 10 mm × 15 mm (FWHM). Further improvements are in progress to enhance the laser power and efficiency of beam transportation to the μSR spectrometer, where a final beam size of a few millimeters is expected. After completion of the initial commissioning phase, experiments will be conducted in area U1A.

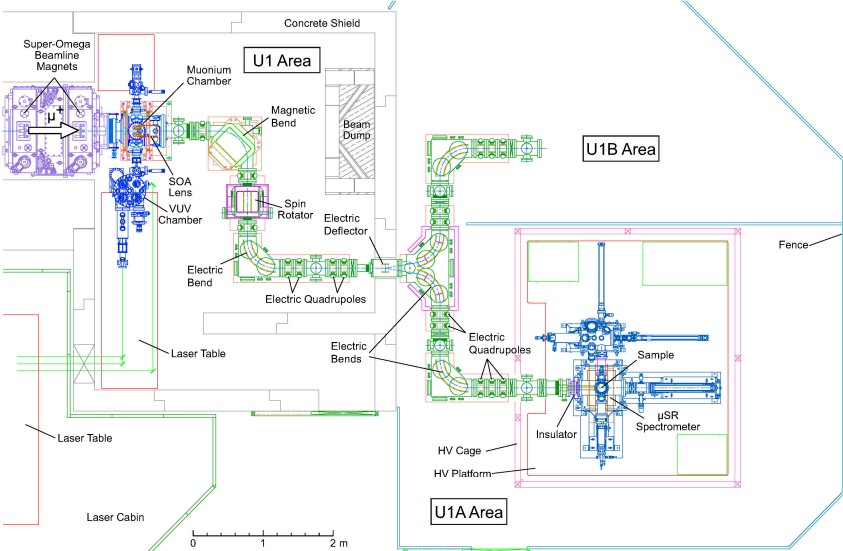

**Figure 8.** Layout of the ultra-slow muon (USM) beamline. A high-flux surface muon beam is injected into a hot W target in the muonium chamber (left), from which thermal muonium atoms evaporate into a vacuum. These muonium atoms are ionized by a laser system and accelerated and focused by an electrostatic lens for transport at 30 keV. The transport beamline consists of one magnet bend, three electric bends, and electric quadrupoles, and it is electrostatically isolated from the high-voltage cage accommodating the muon spin rotation (μSR) spectrometer.

### 3.3. S-Line

The S-line, which is located in experimental hall No. 1 of the MLF building, is designed to provide surface-muon beams (with momenta of 28 MeV/c, which are called "slow" muon beams) for conventional μSR experiments simultaneously performed in the four experimental areas. The beamline is designed to explore the potential advantages of high-flux surface muons in condensed matter physics, and it is typically expected that it will facilitate the investigation of smaller samples and/or require shorter data acquisition (DAQ) periods. In particular, the latter characteristic enables stroboscopic observations of time-evolving phenomena to be performed. While the beamline will eventually serve four experimental areas (S1–S4, as indicated in Figure 9), so far only the branch to area S1 has been completed. The construction work performed thus far includes that for the power supply yard and concrete shield, which were completed in 2013, followed by the installation of beamline magnets and other components in 2014. The first delivery of surface muons to area S1 was confirmed by time-of-flight measurements conducted during the beamline commissioning performed in 2015.

The basic concept of the S-line is similar to that of the D-line, which enabled duplicate design of some components, such as the vacuum chamber and DC separator, including the high-voltage electrodes and correction magnets that are currently used in the D-line. Meanwhile, a Cockcroft–Walton-type high-voltage generator, which had served the U-line DC separators and proved to be more stable than the corresponding device in the D-line, was adopted to achieve stable beam operation. Furthermore, the DC separator is located upstream from the second bending magnet, unlike in the D-line. It is expected that the drift length between the DC separator and the beam slits, which is longer than that in the D-line, will provide improved separation and reduce positron contamination.

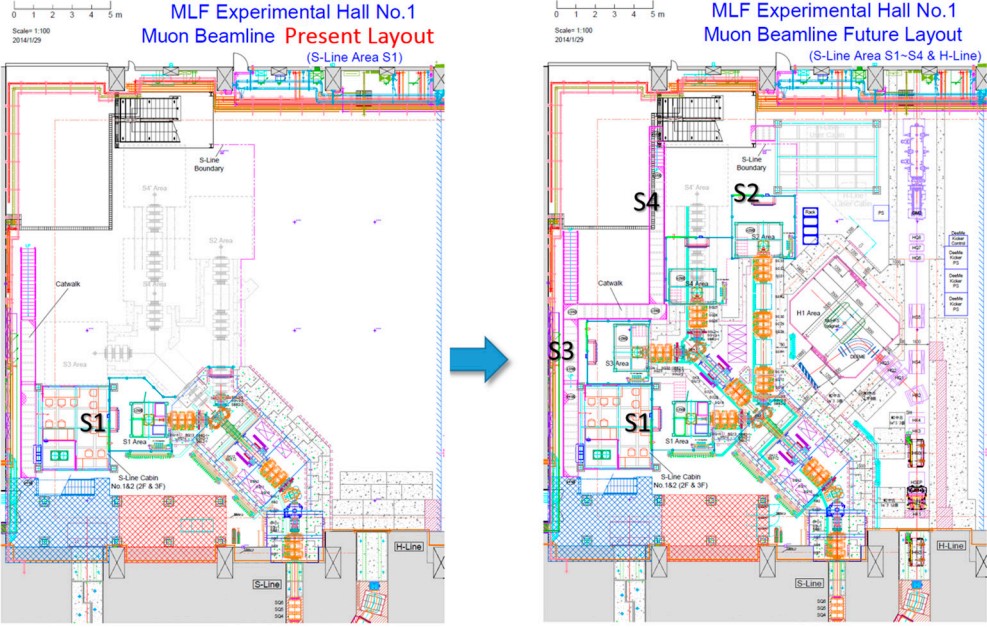

**Figure 9.** Current layout of the S-line to experimental area S1 (**left**) and a provisional floor plan for the completed S-line complex with four branches, one to each of the experimental areas (**right**). The cross-hatched areas near the shielding wall to the M1/M2 tunnel indicate the power supply yards.

The main difference between the S- and D-lines is that an *electric* kicker system is used to deliver single-pulsed muons to area S1 (and to areas S2–S4 in the future). In particular, symmetric operation to distribute two pulsed beams to areas S1 and S2 within a switching time of less than 300 ns is realized using bipolar high-voltage power supplies based on solid-state Marx generators (Figure 10). In addition, a window-frame-type switchyard magnet is included in the kicker chamber to utilize

double-pulsed muon beams. It should be noted that the influence of leakage fields from the septum magnet to the beam orbit toward areas S3 and S4 is not negligible. To minimize this influence, magnetic stainless steel is employed for the septum vacuum chamber. Every power supply for the beamline magnets is equipped with a polarity changer, so that negative muons (cloud muons) can be transported to area S1.

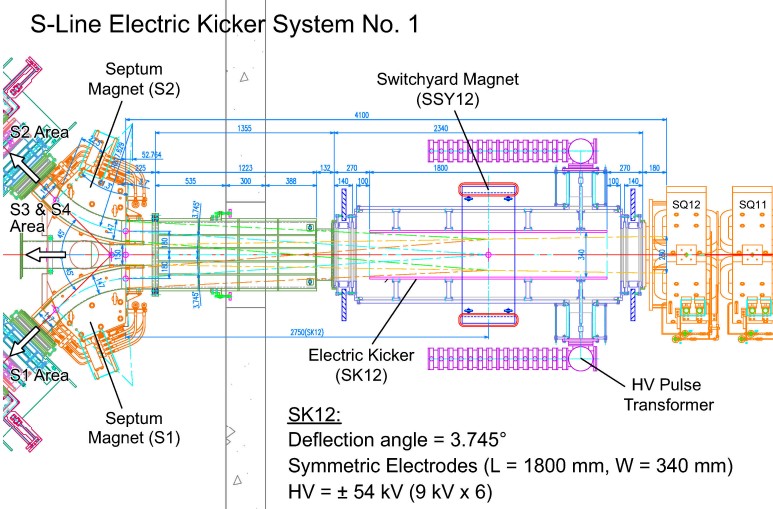

**Figure 10.** Layout of the S-line electric kicker system that delivers single-pulsed muon beam to areas S1 and S2. The muon beam can also be sent straight to areas S3 and S4, where installation of a similar kicker system is planned.

Since the first beam delivery to area S1, beamline commissioning has been intensively performed using the μSR spectrometer, ARTEMIS (the Advanced Research Targeted Experimental Muon Instrument at S-line; see Section 4.1), sponsored by the Element Strategy Initiative Project for Electronic Materials. A custom-made automatic beam-tuning program is used to obtain a well-focused muon beam at the sample position by monitoring μ-e decay events. A typical surface muon flux of $8.5 \times 10^4$/s for a tailored beam spot size of ⌀20 mm at the sample position has been obtained at 150 kW operation. The commissioning processes for the spectrometer itself and the associated sample environments, such as those related to cryostat operation, sample temperature logging, and unmanned measurements, including DAQ control, are performed in parallel. The IROHA2 framework, which is the standard protocol at MLF for communicating among devices to control sample environments, went online in area D1 soon after a brief test in area S1. Not only the software, but also the hardware, such as the water-cooled thermal shield used to stabilize the detector temperature, was exported after testing in area S1. It should be stressed that area S1 plays an important role in the seamless upgrading of the μSR equipment.

*3.4. H-Line*

While the H-line was originally designed to produce a "high-momentum" muon beam, the experiments currently under consideration rather require high muon flux as well as momentum tunability [28–30], which will also be important for future experiments. To meet these demands, a new beamline optics concept involving a large-aperture muon-capture solenoid, wide-gap bending magnet, and pair of two solenoid magnets with oppositely directed fields was proposed. For the detailed design of the beam optics, however, conventional matrix calculations are not applicable due to the failure of the near-axis approximation for a large-aperture solenoid. Thus, Monte-Carlo particle tracking simulation code, G4BEAMLINE [31], was applied to optimize the beamline magnet parameters. Figure 11 depicts typical surface-muon beam transport results.

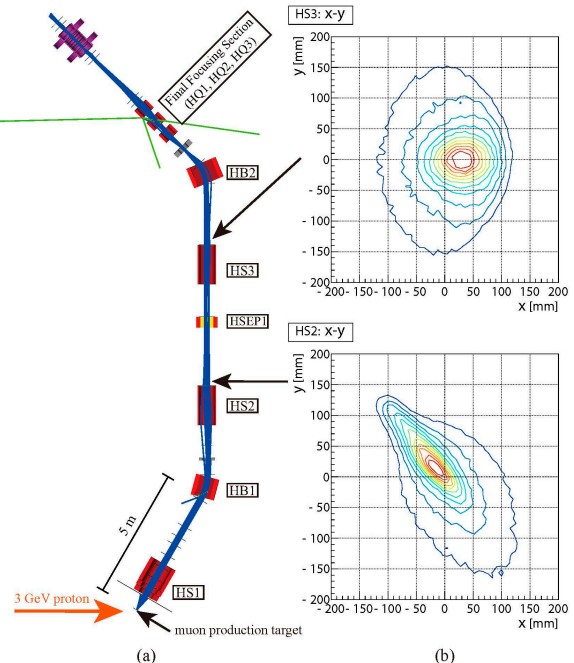

**Figure 11.** (**a**) Typical results obtained by the Monte-Carlo particle tracking simulation code, G4BEAMLINE. The blue lines are the beam trajectories. A surface muon beam from a point source is transported to the first experimental area through three solenoid magnets, HS1, HS2 and HS3, and two bending magnets, HB1 and HB2. The Wien filter is also depicted, although it is not in use in this transmission. (**b**) The beam profiles at the exits of the second and third solenoid magnets, HS2 and HS3, are provided.

The front-end part of the M1/M2 proton beam tunnel has been completed, and the remaining downstream part is under construction, which should enable early completion of J-PARC Phase 1 and will provide beams for two experiments, i.e., high-precision measurement of the hyperfine splitting of muonium [28] and searches for muon-to-electron conversion [29]. In Phase 2, the H-line will be extended beyond the east wall of the MLF building to accommodate a muon beam accelerator with a USM source by employing a technique similar to that used in the U-line and will eventually be utilized for precision measurements of the anomalous magnetic moment of muons [30] and transmission muon microscopy.

The expected total surface-muon flux for the H-line is about $10^8$/s. The high flux attainable using large-aperture magnets has a trade-off relationship with duct-streaming neutrons, which would be more serious in the H-line than in the other muon beamlines. The radiation shield and interlock devices are designed to ensure safe operation against such background radiation.

## 4. Muon Experiment Instruments

### 4.1. µSR Spectrometers

µSR is an experimental technique that is used to measure the time evolution of muon spin polarization in condensed matter. The µSR spectrometer consists of detectors for positrons/electrons emitted by muons, whose spatial asymmetry carries information about the muon spin polarization; electronic devices to measure the time between the muon entry and positron decay times; and an apparatus to control the sample environment characteristics, such as the magnetic field and temperature. In this section, the detectors and electronic devices in the common µSR spectrometer structure are described first and are followed by details about the sample environment apparatus.

It is difficult to apply µSR to pulsed muon sources, such as J-PARC, due to the enormous rate of positron decay at each beam pulse. As explained in the Section 1.3, J-PARC provides the highest flux of pulsed muons in the world; one pulse contains ~$10^5$ muons, which decay with a mean lifetime of 2.2 µs. Thus, the instantaneous positron count rate can be as high as 100 Gcps. Since the typical response time of a fast positron counter (plastic scintillator) is 10 ns, the positron counters must be divided into thousands of channels of small scintillators so that each individual detector measures a reduced count rate on the order of 100 Mcps to handle such high event rates. Thus, a pulsed µSR spectrometer must contain thousands of independent detectors, whose time histograms must be stored independently. To this end, we developed a scalable many-channel positron detector system called Kalliope for use in µSR experiments.

Kalliope (KEK Advanced Linear and Logic-board Integrated Optical detectors for Positrons and Electrons) is an all-in-one detector system for time differential measurement. One Kalliope detector unit is depicted in Figure 12 and consists of 32 channels of individual positron detectors. The time differences between the common start trigger pulse and the multiple hit events on the detectors are recorded by a time-to-digital converter with 1-ns timing resolution and stored in memory on a digital board for subsequent transfer to the DAQ PC via an Ethernet cable using TCP/IP (Transmission Control Protocol/Internet Protocol). The amplifier and discriminator for signals from plastic scintillators (with avalanche photo-diodes (APDs) for photon detection) with a digitally controlled threshold level are realized in an integrated circuit on the analog board, which is configured by the firmware on the digital board. Although Kalliope is compact, it contains everything necessary to measure decay positrons from muons; with Kalliope, Ethernet hubs, and a DAQ-PC, one can measure µSR without preparing other electronics, which represents a major advantage over the conventional NIM (Nuclear Instrumentation Module) and CAMAC (Computer Automated Measurement and Control) or VMEbus (Versa Module Europa bus) crate-based electronics.

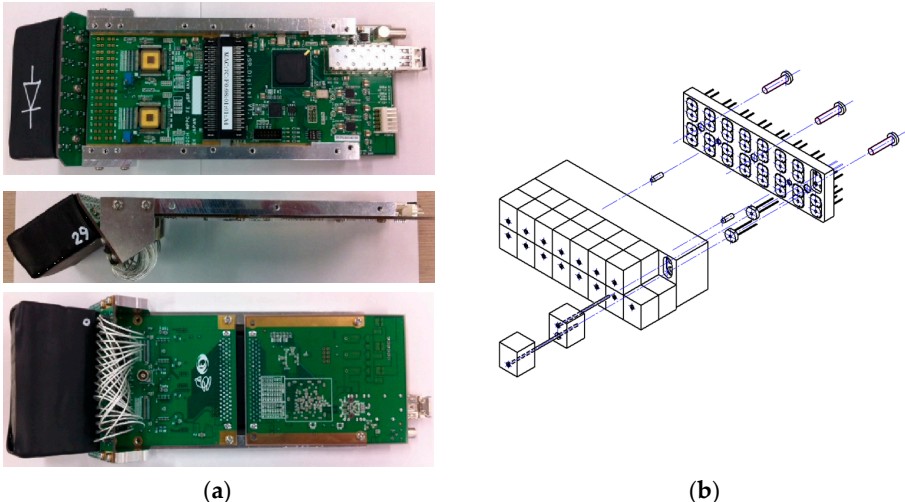

(**a**)              (**b**)

**Figure 12.** (**a**) One Kalliope (KEK Advanced Linear and Logic-board Integrated Optical detectors for Positrons and Electrons) detector, which consists of a scintillator block on a scinti-board, an analog board, and a digital board. Data are transferred to a data acquisition (DAQ) PC via an Ethernet cable. (**b**) A scintillator block consisting of 32 scintillator channels (with dimensions of 10 mm × 10 mm × 12 mm), each with wavelength-shifting fibers and pixel-type avalanche photo diodes (MPPC by Hamamatsu) installed.

Two identical general-purpose µSR spectrometers are installed in areas S1 and D1 (see Figure 13). The spectrometer in area S1 is named ARTEMIS (hereafter denoted as S1-ARTEMIS). Each of these spectrometers contains 40 Kalliope units that cover 21.2% of the total solid angle for positron/electron detection. The detector arrangement is illustrated in Figure 14. The power supply and Ethernet hubs

for the Kalliope units are stored in the base of each spectrometer for portability. The largest Helmholtz coils can apply magnetic fields of up to 0.4 T at the sample position with a DC current of 1000 A. The magnet base has a rotating table with a stopper pin, so that the 0.4 T field can be applied using either longitudinal or horizontal transverse field geometry. Each spectrometer is also equipped with vertical square Helmholtz coils (14 mT for 100 A) and three sets of stray-field compensation coils (1 mT for 10 A) to achieve a zero field at the sample position. The specifications of these spectrometers are summarized in Table 2.

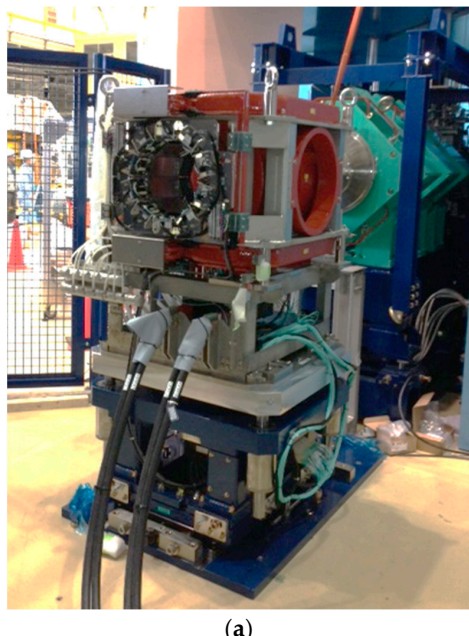
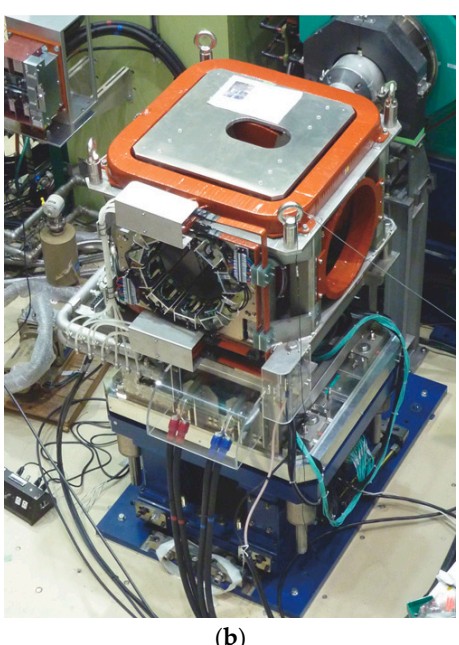

(**a**)                    (**b**)

**Figure 13.** Twin general-purpose μSR spectrometers: (**a**) S1-Advanced Research Targeted Experimental Muon Instrument at S-line (S1-ARTEMIS); and (**b**) the D1 instrument. Each spectrometer includes 40 Kalliope units for positron/electron detection that cover 21.2% of the solid angle. Magnets are equipped to apply magnetic fields of up to 0.4 T and 14 mT in the horizontal and vertical directions, respectively, and xyz stray-field compensation coils are employed to produce a field of 0 T at the sample position.

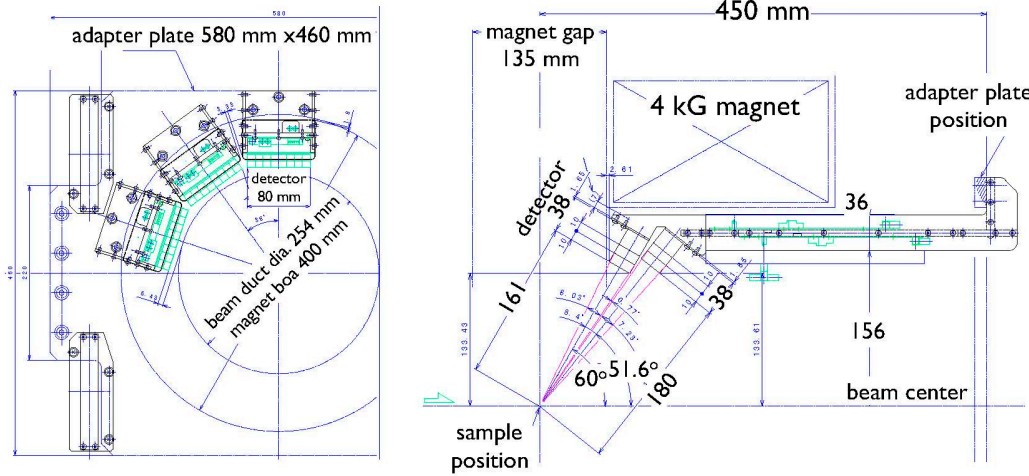

**Figure 14.** Arrangements of the positron/electron counters in the twin general-purpose spectrometers.

**Table 2.** Specifications of the twin μSR spectrometers in areas D1 and S1.

| Magnets | |
| --- | --- |
| Horizontal field $B_z$ (//beam) or $B_x$ ($\perp$beam) | Maximum 0.4 T (1000 A, 100 V) |
| Bore and gap | ø400, 135 mm |
| Homogeneity | 0.1% ø20 mm, 1% ø50 mm |
| Stability | Drift < 0.01% for 8 h Ripple Vp_p < 0.01% |
| Vertical field $B_y$ (($\perp$beam, vertical) | Maximum 14 mT (25 mT with optional coils) (100 A, 40 V) |
| Homogeneity | 0.1% ø20 mm, 1% ø50 mm |
| Stability | Drift < 0.05% Ripple Vp_p < 0.1% |
| Correction field coils used to achieve a zero field (CCx, y, z) | Maximum 1 mT (10 A, 20 V) |
| Homogeneity | 0.5% ø20 mm, 10% ø50 mm |
| Positron Counters | |
| Number of channels | 32 channels × 40 Kalliope units = 1280 channels (640 pairs) |
| Detector solid angle | 21.2% of 4π (2 × 8 array of 10 mm × 12 mm scintillator cubes: covers an area of 1920 mm$^2$ at 180 mm and 161 mm from the sample) |
| Sample Insert Area | |
| Bore and gap | ø254, 135 mm |

Because of the instantaneous high count rate of positrons/electrons in pulsed-μSR measurements, the transient properties of the preamplifiers that are related to the signals from the APDs embedded in the custom analog chip (application-specific integrated circuit, ASIC) are essential to ensure the high performances of the Kalliope units, which significant efforts have been made to improve. The latest ASIC (called "Volume 2014") consists of 100 MHz current amplifiers with a pole-zero cancelation circuit for time constant conversion. Figure 15 presents the example of μSR spectra obtained using S1-ARTEMIS with a count rate of 200 Mhits/h, which corresponds to the rate expected for a 16 mm × 16 mm sample and 1 MW proton beam power.

The computer interface for user operation of the μSR spectrometer is based on the IROHA2 framework developed at MLF, a common user interface for DAQ and device control furnished with auto-run features. The sample environment characteristics, such as the temperature and magnetic field, are set using a PC in the auto-run sequence, and their stability is waited for if necessary before the measurement run starts. The run ends automatically when the preset count is achieved, and the sequence proceeds to the next step. The auto-run progress and current sample environment status are continuously displayed on a web page, so that the user can monitor the experiment remotely via the Internet.

For conventional μSR measurements, various sample environments are available. For the sample temperature, a He-free top-loading dilution refrigerator ($\geq$50 mK), a He gas-flow cryostat (3–400 K), and an infrared furnace ($\leq$1000 K) are ready for use. As mentioned above, magnetic fields can be applied along the beam direction ($\leq$0.4 T) or perpendicular to the beam direction ($\leq$0.025 T). It is also possible to perform μSR measurements under light illumination using a flush lamp synchronized with the muon pulses.

To enable μSR measurements to be performed on small samples ($\geq$5 mm × 5 mm) by reducing the background events from muons that missed the sample, a "fly-past chamber" is used. The fly-past chamber is a long vacuum vessel in which samples are suspended with minimal surrounding equipment for cryogenic control, and muons that missed the sample fly away downstream to prevent the positrons/electrons that they emit from hitting the detectors.

To improve the time resolution, which is limited by the beam pulse width of 80–100 ns, a device called a "muon beam slicer" was once installed in the D-line to area D1 [32]. The beam slicer consists of an electric kicker, a pulsed electric power supply with a fast rise time, and a correction magnet. By applying a pulsed electric field ($\pm$80–100 kV) that is synchronized with the muon pulses, a muon pulse can be sliced to have a narrow structure ($\geq$20 ns) at the expense of reducing the muon flux.

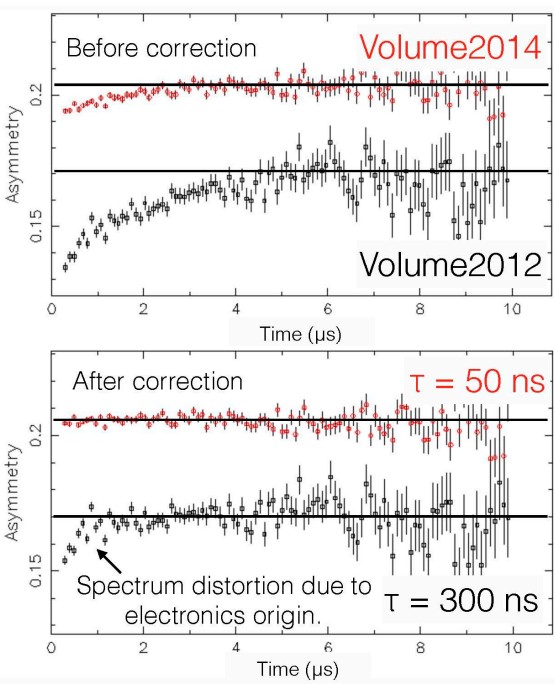

**Figure 15.** μSR time spectra obtained by S1-ARTEMIS with the earlier application-specific integrated circuit (ASIC) ("Volume2012") and the latest ASIC ("Volume2014"). The top and bottom panels respectively depict the spectra before and after pileup correction using the standard formula $N = N_{obs}/(1 - \tau N_{obs})$, where $\tau$ is the dead-time parameter. (The Volume2012 spectra suffer from distortion due to the insufficient transient performances of the voltage amplifiers.)

### 4.2. μSR Spectrometer for USMs

USMs with energies of ~0.2 eV are immediately accelerated to 30 keV for transport to the μSR spectrometer at the end of the USM beamline installed in area U1A (Figure 16). The entire spectrometer assembly, which includes a cryostat, Kalliope positron detectors, magnets, other electronic devices, and their power supplies, is contained within a large metallic cage on an electrically isolated stage. The beam implantation energy can be varied from 0.2 kV to 30 kV by tuning the electrostatic potential of the stage to decelerate USMs in the entry section of the spectrometer. The corresponding muon implantation depth range is 0–200 nm for Cu, where the specimen properties can be investigated continuously from near the surface to the bulk region.

The required features of the USM-μSR spectrometer are identical to those of conventional spectrometers, except that the sample environment that must be compatible with ultra-high vacuum conditions. Meanwhile, care must be taken to minimize the thermal radiation to control the sample temperature without beam windows or radiation shields, which would interrupt the USM implantation. The spectrometer is furnished with a set of Helmholtz coils for applying external magnetic fields of up to 0.14 T along the beam direction. Since a muon spin rotator (Wien filter) is installed upstream from the spectrometer along the USM beamline to modify the initial muon spin direction from 0° (along the beam direction) to 90°, one can perform μSR measurements under transverse or longitudinal field conditions using a single set of magnets. To enable sample exchange without requiring the high vacuum to be broken, a load-lock chamber is attached next to the main vacuum chamber of the spectrometer. The load-lock chamber is also used for sample preparation and surface condition characterization.

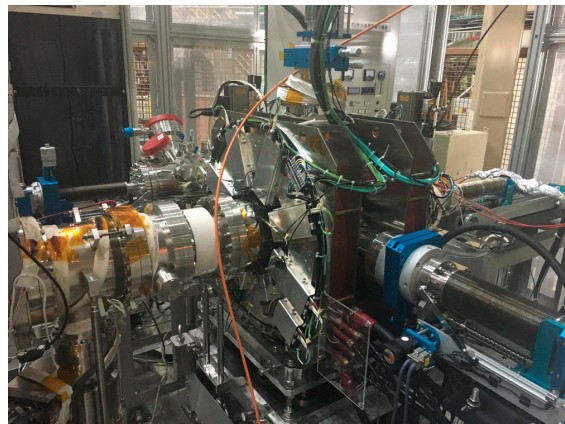

**Figure 16.** μSR spectrometer installed in area U1A for use in USM experiments. The entire assembly is placed in a large cage on a stage electrically isolated from the ground.

## 5. Recent Muon Application Highlights

### 5.1. μSR Studies in Condensed Matter Physics and the Materials Sciences

μSR is one of the most natural muon applications in that it utilizes the sensitivity of magnetic fields to explore the electronic properties of matter via implanted muons. As implanted "pseudo-hydrogen," muons also provide unique opportunities to probe the simulated state of interstitial H via the muon-electron hyperfine parameters. Because of the negligibly small difference (~0.5%) between the reduced electron masses of H and neutral muonium ($Mu^0$, the muonic analogue of a neutral $H^0$ atom), the electronic state of implanted $Mu^0$ is mostly equivalent to that of interstitial $H^0$.

As mentioned in Section 4.1, areas D1 and S1 in MUSE are currently furnished with general-purpose μSR spectrometers, whose abilities to be used for conventional μSR measurements are almost identical. Although the sample environment characteristics, e.g., the accessible temperature and magnetic field ranges, are limited so far, these instruments have proven the usefulness of μSR for the investigation of the local electronic properties of a variety of materials.

One such example is the identification of a new antiferromagnetic phase in a prototype Fe-based superconductor, $LaFeAsO_{1-x}H_x$ (LFAO-H), over a range of unprecedentedly high carrier concentrations [33]. It was previously reported that the range of carrier doping could be extended to much higher concentrations ($x > 0.2$) by substituting O with H instead of F, and the secondary maximum of the superconducting transition temperature ($T_c$) was observed with $x \sim 0.35$ [34]. The results of a preliminary nuclear magnetic resonance (NMR) study suggested the emergence of a certain kind of magnetism in the overdoped region [35]. Based on this finding, a team of experts on muons, neutrons, and synchrotron radiation (SR)-X-rays (working together for the "Element Strategy Initiative" project being conducted at the Condensed Matter Research Center) has made a concerted effort with a group from Tokyo Institute of Technology to elucidate the electronic properties of LFAO-H with large H concentrations. They employed the muon spin rotation technique to map out the dependence of the Néel temperature ($T_N$) on $x$ in a timely fashion (Figure 17). Subsequent neutron and SR-X-ray measurements indicated that the magnetic structure is different when $x$ is high compared to when it is low and that the magnetic transition accompanies a structural change to a non-centrosymmetric structure (Aem2) at $x = 0.5$. These observations suggest that the new phase might be regarded as another "parent" for the secondary superconducting phase.

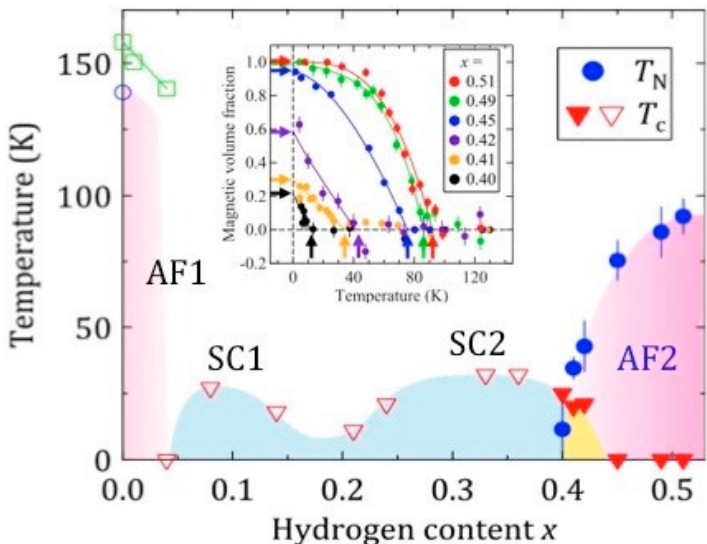

**Figure 17.** New magnetic phase (AF2), for which the Néel temperature ($T_N$) was mapped out by performing μSR measurements. Inset: Magnetic volume fraction vs. temperature determined via μSR [33]. $T_c$: superconducting transition temperature.

Yet another example is the observation of a peculiar magnetic ground state in an Ir spinel compound. Geometrical frustration in electronic degrees of freedom, such as spin, charge, and orbit, which is often realized in stages in highly symmetric crystals, has been an important topic in condensed matter physics. Inorganic compounds with $AB_2X_4$ spinel structures have provided platforms for the investigation of unusual physical properties related to geometrical frustration. A thiospinel compound, $CuIr_2S_4$, is a recent example of such a compound. In $CuIr_2S_4$, mixed-valent Ir ions form isomorphic octamers, $Ir^{3+}_8S_{24}$ and $Ir^{4+}_8S_{24}$, with lattice dimerization of $Ir^{4+}$ pairs in the latter, upon the occurrence of a metal–insulator (MI) transition at 230 K. While it has been proposed that a pair of $Ir^{4+}$ atoms ($5d^5$, total spin $S = 1/2$) forms a non-magnetic spin-singlet dimer driven by orbital order and associated Peierls instability, the magnetic properties of the ground state remain to be clarified microscopically.

The results of recent muon and Cu-NMR studies have indicated that a spin glass-like *magnetic* ground state can be realized in $CuIr_2S_4$ below ~100 K [36]. As shown in Figure 18, slow Gaussian damping was observed at 200 K, which is expected for muons exposed to random local magnetic fields from nuclear magnetic moments (primarily from $^{63}Cu$ and $^{65}Cu$ nuclei in $CuIr_2S_4$), indicating that the compound is non-magnetic at this temperature. In contrast, fast exponential depolarization sets at below ~100 K, where the depolarization rate as well as the relative amplitude of the depolarizing component increases with decreasing temperature. These observations contradict the naive expectation based on the currently accepted scenario that $Ir^{4+}$ pairs form non-magnetic spin-singlet states upon MI transition. The spin glass-like behavior suggests that competing interactions influence the $Ir^{4+}$ atoms, leading to a magnetically frustrated state. It has been proposed that the spin-orbit interaction, which has gained rapid recognition in recent years as an important factor in the physics of 5d electron systems, e.g., $Sr_2IrO_4$, might be the origin of this frustration as well as unquenched local spins, where $Ir^{4+}$ is represented as an eigenstate of an effective isospin $J_{eff} = 1/2$ multiplet.

It is known that substitution of Cu with Zn in $CuIr_2S_4$ suppresses the MI transition, eventually leading to superconductivity in $Cu_{1-x}Zn_xIr_2S_4$ for $x > 0.25$. A μSR study of Zn-substituted samples indicated that the spin glass-like magnetism was strongly suppressed compared with that in $CuIr_2S_4$ (see Figure 18c,d), which parallels the characteristics of high-$T_c$ cuprates and/or Fe-pnictides. Thus, $CuIr_2S_4$ may serve as a new platform for the study of superconductivity under the strong influence of the spin-orbit interaction.

Meanwhile, muons are used to simulate the electronic structure of interstitial H in materials in which H may play a significant role. One related topic that recently gained prominence was the study of the electronic properties of barium titanate ($BaTiO_3$). The perovskite oxide $BaTiO_3$ is one of the most important ferroelectric materials that is widely used in electronic devices. It exhibits ferroelectricity at ambient temperature, and the associated high dielectric constant is indispensable for downsizing multilayer ceramic capacitors.

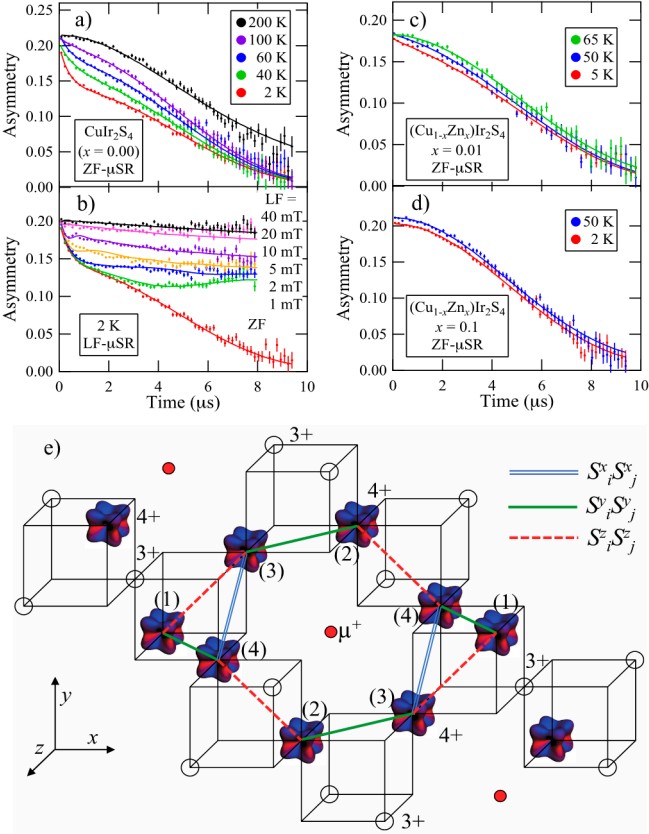

**Figure 18.** (**a**) Time-dependent µSR spectra observed at several temperatures in a powder sample of $CuIr_2S_4$ under zero external field. (**b**) µSR spectra at 2 K under various longitudinal fields. µSR spectra in $Cu_{1-x}Zn_xIr_2S_4$ under zero external field with Zn contents of (**c**) x = 0.01 and (**d**) x = 0.1. (**e**) Octamer configuration associated with charge order in $CuIr_2S_4$. The exchange interaction $S^y_i S^y_j$ for each $Ir^{4+}$ pair is shown by a line along the relevant $\gamma\gamma$ bond ($\gamma$ = x, y, z). The octupolar manifold at each $Ir^{4+}$ site represents the spin density profile in a hole with an isospin-up state (an eigenstate of the $J_{eff}$ = 1/2 multiplet under strong spin-orbit interaction). The bond lengths of the (1)–(4) and (2)–(3) $Ir^{4+}$ pairs are reported to shrink by 15% upon charge ordering and the associated structural transition [36].

Infrared absorption spectroscopy results have indicated that H impurities in $BaTiO_3$ may form O–H bonds, suggesting that H can be stabilized as interstitial $H^+$ to form $OH^-$ ions. First-principles calculation results have further suggested that the electronic levels associated with the $OH^-$ state may not be formed in the band gap, remaining near the bottom of the conduction band so that it can serve as a shallow electron donor. The carrier doping by $OH^-$ formation may cause a serious problem, specifically, that the performance of $BaTiO_3$ as an insulating material for capacitors could be degraded by H impurities in the environment.

A µSR experiment was conducted to test this possibility using muons to simulate the electronic state of interstitial H in matter [37]. Satellite signals were observed around the center line (corresponding to the $\mu^+$ state) in the µSR spectra measured in $BaTiO_3$ at low temperatures, which is a typical sign that $Mu^0$ with an extremely small hyperfine parameter is formed (see Figure 19a).

The Mu$^0$ state was also found to disappear upon warming up to a temperature above 100 K, which is interpreted as "ionization" of Mu$^0$ occurring with a small activation energy of ~10 meV (see Figure 19b). Thus, muonium has been demonstrated to act as a shallow donor in BaTiO$_3$, strongly suggesting that interstitial H would exhibit similar behavior.

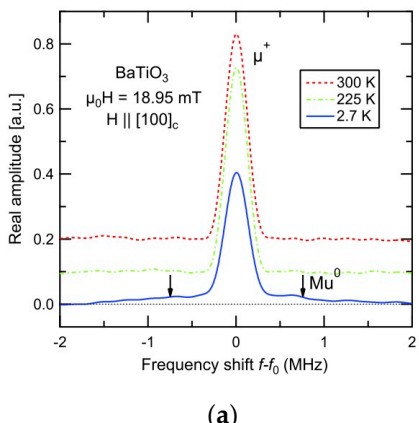
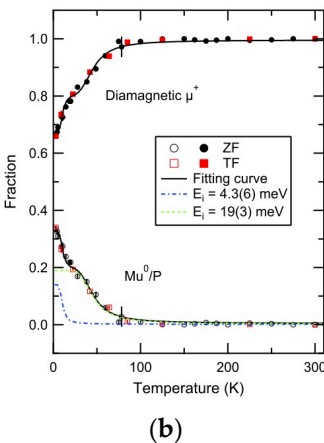

(**a**)    (**b**)

**Figure 19.** (**a**) Fourier transform of the μSR spectrum of BaTiO$_3$. (**b**) Temperature dependences of the fractional yields of μ$^+$ and Mu$^0$ [37].

## 5.2. Non-Invasive Element Analysis

When a negative muon is implanted into matter, it loses kinetic energy, is captured by the Coulomb field of the nucleus, and forms a muonic atom consisting of the nucleus and a negatively charged muon in place of an electron. As illustrated in Figure 20, upon muonic atom formation in a highly excited level, such as that with a principal quantum number of ~15, the captured muon immediately de-excites to lower muon atomic levels by emitting characteristic muonic X-rays with energies unique to each element in the matter. This feature enables the use of muonic X-rays for non-destructive element analysis.

Because the mass of a muon is large (207 times that of an electron), muonic X-rays are about 200 times harder (i.e., higher in energy) than electronic X-rays from the same element. For Cu, for instance, the muonic Kα X-ray has an energy of about 1500 keV, whereas that of the electronic X-ray is only 8 keV. While the latter is blocked by 100-μm-thick Cu foil, muonic X-rays can easily penetrate a Cu plate several millimeters thick. Thus, one of the major advantages of muonic X-rays over electronic X-rays is that they can be used for element analysis deep inside a bulk specimen. This capability is particularly useful for light elements, since conventional fluorescent X-ray analysis is insensitive to elements lighter than Na due to the limited penetration depths of X-rays. An example of a muonic X-ray spectrum obtained for a meteorite specimen is shown in Figure 20, where signals from light elements are detected through a wall of glass tube [38].

Moreover, since tuning the beamline parameters can vary the implantation energy of negative muons, they can be stopped at any depth from a few micrometers to several centimeters. Therefore, muonic X-rays provide a unique means of 3D bulk-elemental analysis when combined with a 2D X-ray imaging device or beam-scanning technique.

As mentioned in Section 3.1 the negative muons delivered at the D-line originate from π$^-$ decays in flight in a long superconducting solenoid magnet. Although the beamline was originally designed to transport muons with energy of up to ~165 MeV (corresponding to a momentum of 250 MeV/c), the specifications of the current beamline components only enable the transport of muons with momenta of up to 120 MeV/c. Another boundary on the low-energy side was caused by the beam windows of the superconducting solenoid that are placed at both ends to protect the vacuum and low temperature within the cold bore.

Recently, we installed a new superconducting solenoid containing a warm bore with an inner radius of ø20 mm without beam windows so that negative muons with extremely low momenta can be delivered to area D2. Beam commissioning is currently in progress to attain a relatively high flux and narrow momentum width to expand the scope of muonic X-ray element analysis using low-energy negative muons.

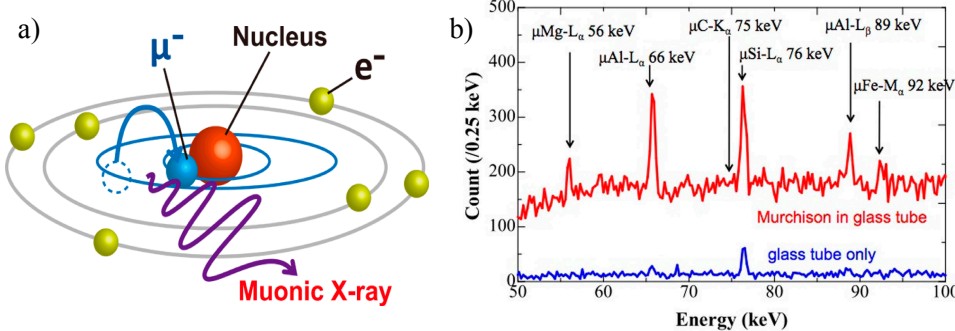

**Figure 20.** (**a**) Schematic illustration of the formation of a muonic atom, an atomic system that has a negatively charged muon in place of an electron. The element-specific X-ray is emitted upon successive transitions of muons to the lower-energy muonic orbitals. (**b**) Muonic X-ray spectra from the powdered Murchison meteorite in a $SiO_2$ glass tube. A clear signal of Mg and a marginally resolved signal of Fe from the sample were detected through the 1-mm thick $SiO_2$ glass wall [38].

**Acknowledgments:** We would like to thank all of the members of the MUSE construction team and collaborators, in particular, Jack L. Beveridge, Jaap Doornbos, Gerd Heidenreich, Yoshiro Irie, Yasuhiro Makida, Kazutaka Nakahara, Toru Ogitsu, Naohito Saito, and Soshi Takeshita, for their significant contributions during the early stage. Thanks also to the present and past MUSE staff, including Taihei Adachi, Hiroshi Fujimori, Yutaka Ikedo, Takashi U. Ito, Yasuo Kobayashi, Jumpei Nakamura, Kanetada Nagamine, Takashi Nagatomo, Yu Oishi, and Amba D. Pant, for their crucial contributions in the development of the MUSE facility. We also would like to express our sincere gratitude to Shoji Nagamiya and Yujiro Ikeda for the kind support provided by J-PARC Center under their directorship. Finally, we dedicate this article as a special tribute to the late Kusuo Nishiyama, who devoted his entire life to bringing MUSE into existence.

**Author Contributions:** The draft of Section 1 was prepared by Y.M., Section 2 by S.Mak. and S. Mat, Section 3 by K.S., P.S., A.K., and N.K., Section 4 by K.M.K and W.H., Section 5 by R.K. and Y.M., respectively according to their expertizes and primary fields of contribution to the MUSE facility. Revision of the entire draft for integration into the final production was made by R.K.

**Conflicts of Interest:** The authors declare no conflict of interest.

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
