# Peer review of "Materials and Life Science Experimental Facility at the Japan Proton Accelerator Research Complex IV: The Muon Facility"

_qubs, doi:10.3390/qubs1010011_

Round 1

Reviewer 1 Report

This is a very timely report on the status of the MUSE facility at JPARC and should be well received by the community.

However,iIt would be useful to the interested readersto be able to expand some of the figures to discern somewhat more detail in the information.  It seems that most of the figures are of .jpeg format, probably imported into a .docx (or similar) program perhaps originally as pdf figures, and then re-exported to pdf. This is often problematical and first exporting the figures as .png (or similar) prior to incorporating them into the .docx may assist the the final figure quality.

Author Response

We prepared revised most of the quoted figures with improved resolution. (Those in the draft were only for the anonimous review.)  All the figures will be provided in the eps form for the final production.

Reviewer 2 Report

This is an interesting review about the muSR activities at J-PARC.

The manuscript merits a proper publication, but some details have to be adressed first.

1) The authors should be careful to stay consequent throughout the manuscript. Numbers for the same parameters are sometimes different in different section. For example: power deposited in target (3.5 kW or 3.9 kW), proton beam speared in target (5% or 6.5%), transverse field for the spectrometers (0.14 T or 0.014 T), the angle of the beamlines compare to the target (45 degrees or 135 degrees) etc...

This is an important point for the readers

2) The quality of the figures has to be improved. Please use vector images. The exprimental Hall in figure 1 is microscopic and one does not get where are for example the M1 and M2 tunnels. The same for figure 6.

A single figure for the D-line is missing (the U and S lines have a figure)

3) In the introduction and throughout the papers, the authors should also specify the size of the beam at the sample position. To give a simple rate is not acceptable.

4) Also a reader understands only through the lines that the operation is done with a 20mm target (for example in Section 1.1).

5) The table 1 is not clear concerning the last 2 lines. What do they represent and why there is such a big difference between simulation and measurements.

6) The authors should define the HIP technique

7) Please provide a reference for the "ForTune" system

8) The authors give the impression that they already accomplished experiments using ultra-slow muons (USM). Is that true? Again in Section 3.2 the position of the detection of the USMs should be given and also the size of the beam.

9) I am also missing comparison with other facilities (please no extrapolations!). How the present numbers compare with other facilities?

So, to conclude: good paper but still some work needed.

Author Response

We thank detailed comments for the referee.  We made extensive revision to sort out some inconsistencies among sections. Figures were also revised, and they are provided in the eps format. Coming to the beam spot size and muon flux, we included some numbers for the readers convenience. However, we find it difficult (or unfair) to compare these with other facilities without detailed specification on the conditions of evaluation. We believe that other criticisms were met in the revised manuscript.

Quantum Beam Sci. EISSN 2412-382X Published by MDPI AG, Basel, Switzerland RSS E-Mail Table of Contents Alert
Back to Top